# Recent Advances in the Gastrointestinal Complex in Vitro Model for ADME Studies

**DOI:** 10.3390/pharmaceutics16010037

**Published:** 2023-12-27

**Authors:** Kazuyoshi Michiba, Kengo Watanabe, Tomoki Imaoka, Daisuke Nakai

**Affiliations:** Drug Metabolism & Pharmacokinetics Research Laboratory, Daiichi Sankyo Co., Ltd., 1-2-58, Hiromachi, Shinagawa-ku, Tokyo 140-8710, Japan; watanabe.kengo.dy@daiichisankyo.co.jp (K.W.); imaoka.tomoki.hd@daiichisankyo.co.jp (T.I.); nakai.daisuke.jf@daiichisankyo.co.jp (D.N.)

**Keywords:** gastrointestinal absorption, complex in vitro model, microphysiological system, Ussing chamber, organoid, induced pluripotent stem cell

## Abstract

Intestinal absorption is a complex process involving the permeability of the epithelial barrier, efflux transporter activity, and intestinal metabolism. Identifying the key factors that govern intestinal absorption for each investigational drug is crucial. To assess and predict intestinal absorption in humans, it is necessary to leverage appropriate in vitro systems. Traditionally, Caco-2 monolayer systems and intestinal Ussing chamber studies have been considered the ‘gold standard’ for studying intestinal absorption. However, these methods have limitations that hinder their universal use in drug discovery and development. Recently, there has been an increasing number of reports on complex in vitro models (CIVMs) using human intestinal organoids derived from intestinal tissue specimens or iPSC-derived enterocytes plated on 2D or 3D in microphysiological systems. These CIVMs provide a more physiologically relevant representation of key ADME-related proteins compared to conventional in vitro methods. They hold great promise for use in drug discovery and development due to their ability to replicate the expressions and functions of these proteins. This review highlights recent advances in gut CIVMs employing intestinal organoid model systems compared to conventional methods. It is important to note that each CIVM should be tailored to the investigational drug properties and research questions at hand.

## 1. Introduction

The intestine plays a crucial role in determining the systemic exposure and pharmacological response of orally administered drugs [1,2,3]. Intestinal absorption is achieved through coordinated actions of dissolution in luminal fluid, permeability through the mucosal epithelial barrier, efflux transporter escape, and intestinal metabolism. All of those factors act as gatekeepers in the intestine, limiting oral drug absorption, which requires the identification of key factor(s) governing intestinal absorption for each investigational drug and the use of an appropriate in vitro system(s) to assess/predict intestinal absorption in humans.

Conventional cell monolayer systems such as Caco-2 cells (a human epithelial colorectal adenocarcinoma cell line) and MDCKII cells stably transfected with a transporter or metabolic enzyme of interest are widely used to study intestinal absorption kinetics. High similarity with absorptive enterocytes has led Caco-2 cell monolayer systems to be amenable to studying the passive penetration of a wide range of compounds, as a high correlation was observed between the apparent permeability coefficient (P_app_) across Caco-2 cell monolayer systems and in vivo Fa (fraction absorbed in gut) values for passively transported compounds [4]. In addition, Caco-2 cells maintain fairly high expression of major drug transporters mediating xenobiotic uptake, efflux, and secretion, such as P-glycoprotein (P-gp/MDR1), breast cancer resistance protein (BCRP), and organic anion transporting polypeptide (OATP) 2B1 [5,6,7,8], although not exactly the same expression pattern as in the human small intestine [9,10]. Therefore, these cells have been widely accepted as useful models for studying active-transported compounds. 

On the other hand, less attractive results were reported using Caco-2 cells for studying gut metabolism; Caco-2 cells show a much lower expression of phase I drug metabolizing enzymes compared to human duodenum, including cytochrome P450 (CYP) 3A4, CYP2C9, and CYP2C19 [9,11]. More importantly, deficiency of CYP3A4 enzymes is a remarkable drawback to studying Phase I metabolism and the first-pass effect of orally administered drugs, especially when Fg (fraction escaping gut metabolism) significantly contributes to overall intestinal absorption. In order to assess Fg, as alternative methods, human intestinal microsomes or human liver microsomes have been used. Those simple in vitro systems allow for the assessment of intrinsic metabolic clearance in the intestine. However, kinetically speaking, as Fg is affected by intrinsic permeability as well as intrinsic metabolism clearance, human intestinal microsomes or human liver microsomes alone do not allow to estimate Fg directly, and those in vitro models that can be directly used to study permeability and intrinsic metabolism, hence Fa and Fg, have long been sought for. Genetically engineered stably CYP3A4-expressed Caco-2 cells are useful to investigate the intestinal availability of CYP3A substrates [12]; however, there are still differences in expression rates of transporters and metabolic enzymes between Caco-2 cells and the human small intestine, and they are unable to address more complex processes (e.g., interplay of uptake/efflux transporter and metabolism, multi-step metabolism). Even in the case of stably transfected Caco-2 cells or MDCKII cells that overexpress multiple transporters and metabolic enzymes of interest, it is difficult to address comprehensive interactions.

The Ussing chamber system, which utilizes fresh human intestinal tissue, is a powerful experimental system to evaluate intestinal membrane transport and metabolism simultaneously with the direct use of human intestinal tissue. Several studies have demonstrated the usefulness of this system for investigating the intestinal absorption process of drugs in humans, as we will describe later (see Section 2). However, the routine use of this system is limited because of several barriers, especially the limited and irregular availability of “fresh” human intestinal tissue samples (the freshness of the tissue sample has a great impact on assay outcomes). Therefore, there remains a need for in vivo-relevant models that maintain the expressions/functions of ADME-related factors almost the same as in the human small intestine and can be available when needed.

In the last decade, much progress has been made for complex in vitro models (CIVMs) as biomimetic devices that functionally model tissue units [13,14,15]. CIVMs not only accurately recapitulate key aspects of human organ structure but biological functional responses as well, rendering opportunities to leverage CIVMs for drug discovery and development. As for the absorption, distribution, metabolism, and elimination (ADME) sciences field, CIVMs help to evaluate critical ADME parameters in humans and select the best clinical candidates and/or predictions of disposition in humans. Recently, a growing number of reports on gut CIVMs have been made employing human intestinal organoids from intestinal tissue specimens or induced pluripotent stem cell (iPSC)-derived enterocytes plated on two-dimensional (2D) or three-dimensional (3D) culture in microphysiological systems (MPS) [13,16,17,18,19,20,21]. Those gut CIVMs display a more physiologically relevant cell culture platform by successful recapitulation of key features of the intestine with a self-renewing system or introduction of physiomimic flow into the system [17,19,20,22,23,24], thereby providing opportunities for more accurate prediction of intestinal absorption in humans.

While those emerging CIVMs possess advantages over conventional in vitro methods, it is essential to clarify the right context of use (CoU) for each CIVM to use, as there are no CIVMs that perfectly replicate the functions of hundreds of metabolic enzymes and transporters in one in vitro model. More importantly, by identifying what types of ADME research questions exist and how CIVMs can support the work (e.g., whether CIVMs are used for drug screening or human dose predictions for selected clinical candidates), we need to select the right CIVMs that fit the purpose.

This review will highlight recent advances in gut CIVMs employing organoid intestinal model systems with and without MPS and describe how gut CIVMs address research questions in ADME studies as compared with conventional methods.

## 2. The Ussing Chamber System Utilizing Fresh Human Intestinal Tissues

The Ussing chamber system enables the evaluation of the bidirectional transport (i.e., mucosal-to-serosal and serosal-to-mucosal) of test compounds across intestinal tissues mounted between two chambers filled with separate solutions that mimic physiological conditions [25,26,27]. Freshly isolated human intestinal tissue segments can be readily mounted in Ussing chambers for investigations into intestinal permeation for in vivo prediction and the mechanisms of intestinal absorption processes. Several studies have reported that the P_app_ values of drugs with different physicochemical and pharmacokinetic properties in the mucosal-to-serosal direction across human intestinal tissue segments showed a sigmoidal relationship with the reported Fa or FaFg values in humans [28,29,30,31]. The Ussing chamber system successfully replicates the functionality of transporters and metabolic enzymes that play critical roles in facilitating or restricting the intestinal absorption of a wide variety of drugs. The coordinated actions of transporter-mediated transport and metabolism in the intestine can be evaluated using human intestinal tissues mounted in Ussing chambers; P-gp- and BCRP-mediated efflux transport and CYP3A-mediated metabolism have been observed in this system [29,30,31,32,33]. In addition, maintenance of functions of representative intestinal uptake carriers, which facilitate the intestinal absorption of hydrophilic drugs containing similar structures to endogenous substrates, was shown in fresh human jejunal tissues by demonstrating that the mucosal-to-serosal transport of substrate drugs for peptide transporter (PEPT) 1, proton-coupled folate transporter (PCFT), and concentrative nucleoside transporters was inhibited by their respective inhibitors [31]. Furthermore, the mucosal-to-serosal P_app_ values of 19 drugs, including several uptake/efflux transporter substrates across human jejunal tissues, showed a good correlation with reported FaFg values in humans. Although Caco-2 cells tend to underestimate the human intestinal absorption of hydrophilic drugs that are substrates of uptake transporters, such as PEPT1 and nucleoside transporters, due to their low expression levels in Caco-2 cells [34,35], the Ussing chamber system utilizing fresh human intestinal tissues provided an appropriate estimation of the FaFg value of uptake transporter substrate drugs. Therefore, the Ussing chamber system utilizing fresh human intestinal tissues is useful for comprehensively predicting the in vivo human intestinal absorption of drugs, regardless of the involvement of intestinal uptake/efflux transporters.

A major drawback of this system is that excised tissues remain viable for a relatively short period of time. The mucosal-to-serosal transport of substrate drugs for PEPT1 and PCFT across the tissues, which were kept in ice-cold buffer for 4 h without continuous oxygenation before starting the assay, decreased to one-thirds of that across the tissues immediately after excision [31]. Furthermore, the 4 h incubation led to a 3-fold increase in the mucosal-to-serosal transport of lucifer yellow, a paracellular transport marker. These results indicate that the transport activities of these uptake transporters and the integrity of tight junctions were decreased only 4 h after the excision of the tissues. Therefore, it is important for the laboratory where the experiments are performed and the site where the tissues are excised to be close in order to maintain freshness. Additionally, the coefficients of variation of the P_app_ values for lucifer yellow (a paracellular transport marker) and propranolol (a passive transcellular transport marker) showed relatively high inter-experimental variability between donors and replicates [31], indicating the need for experiments using intestinal tissues from multiple donors. Limitations of this system also include the limited and irregular availability of fresh intestinal tissue samples, the limited number of test compounds that can be evaluated in one experiment (8–12 parallel chambers), and the complicated preparation process of intestinal tissues. These limitations preclude the routine use of this system in drug screening. Despite these limitations, the Ussing chamber system utilizing fresh human intestinal tissues remains a valuable tool for predicting intestinal absorption in humans for a selected clinical candidate and investigating the mechanisms of intestinal transport and/or metabolism of drugs in humans.

## 3. Intestinal Organoids/Spheroids-Derived Intestinal Epithelial Cells

Intestinal organoids, self-organized 3D structures consisting of intestinal stem cells (ISCs) and mature cells, have emerged as a cutting-edge in vitro system for the study of intestinal biology. Intestinal organoids can be generated from tissue-derived adult ISCs expressing leucine-rich repeat-containing G-protein-coupled receptor 5 [36,37,38,39] or iPSCs [40,41] when embedded in the extracellular matrix (e.g., Matrigel) and cultured with growth media containing critical factors for the maintenance of ISCs (e.g., Wnt, R-spondin, and Noggin) to mimic the in vivo ISC niche microenvironment. This system allows for the long-term culture of ISCs with the ability to self-renew and differentiate into various mature cell types (e.g., absorptive enterocytes, goblet cells, enteroendocrine cells, and Paneth cells). Intestinal organoids are increasingly used in a wide range of research fields, including the study of disease pathogenesis (e.g., genetic diseases, infectious diseases, and cancers) [42,43,44], precision medicine [45], and regenerative medicine [46], as they better recapitulate the complex features of the intestine compared to conventional models.

In recent years, there have been a growing number of reports on the application of intestinal organoids for investigating the impact of pharmacokinetic-related factors on intestinal drug absorption in humans. Stresser et al. investigated the usefulness of 3D human intestinal organoids as a model for evaluating drug-mediated induction of pharmacokinetic-related genes in the intestine [47]. Human ileal organoids exhibited the induction of CYP3A4 mRNA expression (9-fold) and nifedipine oxidase activity (14-fold) by treatment with rifampicin, a typical pregnane X receptor (PXR) agonist. Testing various activators of PXR, constitutive androstane receptor, and aryl hydrocarbon receptor resulted in induction responses that differed from those observed in hepatocytes, suggesting that 3D human intestinal organoids have the potential to be a physiologically relevant model for evaluating the induction of pharmacokinetic-related genes in the intestine. Although the closed geometry of intestinal organoids limits access to the apical surface of the epithelium, 2D cell monolayers can be generated from dissociated 3D intestinal organoids, which are more suitable for transcellular transport assays [48]. Yamashita et al. demonstrated that 2D cell monolayers derived from human biopsy-derived duodenal organoids, which had a trans-epithelial electrical resistance (TEER) value of approximately 200 Ωcm^2^, showed higher expressions and activities of CYP3A4 and carboxylesterase (CES) 2 compared to Caco-2 cells [49]. Unlike Caco-2 cells, the metabolic activity of CES2 was higher than that of CES1 in human duodenal organoid-derived monolayers, consistent with the predominant expression of CES2 in the human small intestine. Furthermore, the treatment of rifampicin induced the mRNA/protein expressions and activities of CYP3A4 and P-gp in human duodenal organoid-derived monolayers, indicating its applicability to studies on drug-mediated induction of pharmacokinetic-related factors. Human duodenal organoid-derived monolayers exhibited a global gene expression profile more similar to that of the human duodenum compared to Caco-2 cells. 

Likewise, Michiba et al. successfully established 2D cell monolayers derived from 3D human jejunal spheroids generated from surgical human jejunal specimens and conducted comprehensive functional studies of drug-metabolizing enzymes and transporters [21]. The differentiated 2D cell monolayers showed a drastic increase in the mRNA expression levels of major intestinal transporters and drug-metabolizing enzymes compared to those in undifferentiated 3D spheroids and had TEER values of approximately 300 Ωcm^2^. The transport activities of major intestinal uptake/efflux transporters (PEPT1, P-gp, and BCRP) as well as the metabolic activities of a wide variety of drug-metabolizing enzymes, including CYP (CYP3A and CYP2C9), phase II enzymes (uridine 5′-diphospho-glucuronosyltransferase [UGT] 1A), and non-CYP phase I enzymes (CES2), were confirmed in human jejunal spheroid-derived monolayers. The efflux ratios of the tested P-gp and BCRP substrate drugs (fexofenadine, sulfasalazine, and rosuvastatin), except for digoxin, in human jejunal spheroid-derived monolayers were roughly consistent with those in human jejunal tissue in Ussing chambers. The accumulation of experimental data is needed to demonstrate that human intestinal spheroid-derived monolayers are an in vivo-relevant model for transporter substrates. Furthermore, efforts have been made to translate the results of in vitro experiments using human jejunal spheroid-derived monolayers into Fg in humans. Fg values were estimated using two different approaches: (1) based on the ratio of the apical-to-basal permeation clearance in the presence and absence of ketoconazole (a potent CYP3A inhibitor), and (2) based on the ratio of the total amount of parent compounds transferred to the basal compartment and the total amount of produced metabolites. Both methods gave predictions of Fg that were well-correlated with the reported in vivo human Fg values for five CYP3A (non-P-gp) substrate drugs. This is significant in that conventional Caco-2 models fail to evaluate CYP3A-mediated metabolism due to the lack of CYP3A expression, highlighting the advantages of human jejunal spheroid-derived monolayers in investigating the relative importance of CYP3A in the overall intestinal absorption of substrate drugs in humans. Moreover, human intestinal microsomes can be used to quantify CYP3A-mediated metabolism; however, it is difficult to evaluate the extent to which metabolism contributes to intestinal absorption because both membrane permeability and metabolism cannot be evaluated simultaneously. The human intestinal organoids/spheroids-derived monolayer system has great potential to investigate the intestinal absorption of substrate drugs by multiple transporters and/or drug-metabolizing enzymes (e.g., the interplay of transporter and metabolic enzyme), although further studies are needed to demonstrate the usefulness of this system. Additionally, Inui et al. successfully established MDR1-knockout intestinal organoids using the clustered regularly interspaced short palindromic repeats (CRISPR)/CRISPR-associated protein 9 system [50]. Specific transporter- or metabolic enzyme-knockout intestinal organoids would serve as useful tools to directly investigate the relative contribution of each isoform to the overall intestinal absorption without the use of inhibitors, which might cause off-target inhibition or have low inhibition potency.

One of the unique properties of the tissue-derived intestinal organoid culture system is that organoids established from adult ISCs isolated from different regions of the intestine (e.g., the duodenum, jejunum, ileum, and colon) retain the global gene expression of the corresponding tissue region [51,52,53,54]. The expression levels of several intestinal transporters are not homogeneous along the intestinal tract. For example, the apical sodium-dependent bile acid transporter (ASBT), which plays a major role in the intestinal (re-)absorption of biliary-secreted bile acids, is predominantly expressed in the lower small intestine (i.e., the ileum) [55,56,57]. However, it is difficult to address regional differences in transporter-mediated membrane transport in the intestine using conventional Caco-2 cells. Additionally, with respect to human iPSC-derived intestinal epithelial cells, it is unclear which part of the small intestine they reflect. In contrast, Middendrop et al. demonstrated that the regional identity of their origin is intrinsically programmed within adult ISCs in the intestinal organoid culture system [51]. Michiba et al. investigated whether the functional expressions of region-specific intestinal transporters could be maintained in human intestinal spheroid-derived monolayers originating from the proximal jejunum and terminal ileum (Michiba et al., in preparation). Consistent with the region-specific expressions in human proximal jejunal and terminal ileal tissues, the expressions of *SLC10A2* (ASBT) and *SLC46A1* (PCFT) were highly restricted to human terminal ileal spheroid-derived monolayers and human proximal spheroid-derived monolayers, respectively. Furthermore, sodium-dependent uptake of [^3^H]-taurocholic acid (an ASBT substrate), which was inhibited by the addition of excess non-radiolabeled taurocholic acid, was observed only in human terminal ileal spheroid-derived monolayers. In contrast, pH-dependent and folate-inhibitable uptake of methotrexate (a PCFT substrate) was observed only in human proximal spheroid-derived monolayers. Therefore, intestinal organoid culture system has great potential to investigate the impact of the region-specific functions of pharmacokinetic-related factors on drug transport and metabolism in the intestine.

Furthermore, recent studies have reported the development of intestine MPS using human intestinal organoids. Kasendra et al. successfully established a primary human Small Intestine Chip (Intestine Chip) that recapitulated intestinal tissue structural and functional features by combining a microfluidic device (2-channel Organs-on-Chips) with human duodenal organoids derived from biopsies [19,20]. The device enables the complex physiologically relevant microenvironment of the human intestine, such as fluid flow and peristalsis-like mechanical motions. The polarized epithelial cells in the Duodenum Intestine Chip exhibited the formation of villi-like structures, differentiation into multiple mature cell types, brush-border digestive enzyme activity, and mucus secretion [19]. They also showed that the Intestine Chip developed a functional intestinal epithelial barrier over a period of multiple days by demonstrating that the P_app_ values of lucifer yellow were maintained at approximately 1 × 10^−6^ cm/s at least until 12 days of culture. Shin et al. discovered that the removal of the secreted Dickkopf-1 (Wnt antagonist) by fluid flow in the basolateral microchannel promoted the formation of villi-like structures in the Intestine Chip [58]. In addition, Kasendra et al. demonstrated that Duodenum Intestine Chip showed appropriate localization of intestinal drug transporters (P-gp, BCRP, and PEPT1) and P-gp efflux activity [20]. The expression of major intestinal transporters such as P-gp, BCRP, PEPT1, OATP2B1, and organic cation transporter 1 in Duodenum Intestine Chip was closer to that in the duodenum compared to Caco-2 Intestine Chip. The expression levels of CYP3A4, PXR, and vitamin D receptor in Duodenum Intestine Chip were significantly higher and much closer to those in the duodenum compared to Caco-2 Intestine Chip. The treatment of rifampicin and 1, 25-dihydroxyvitamin D_3_ resulted in the induction of the protein expression and metabolic activity (metabolism of testosterone to 6β-hydroxytestosterone) of CYP3A4 in Duodenum Intestine Chip. Attention should be paid to the sorption of compounds into the material of microfluidic devices. Wang et al. showed the marked adsorption and absorption of hydrophobic compounds such as midazolam into the polydimethylsiloxane (PDMS)-based microfluidic devices, which are commonly used in the commercially available MPS platforms [59]. They also reported that perfluoropolyether (PFPE)-based devices showed much stronger chemical repellency than PDMS-based devices by demonstrating that the sorption of hydrophobic drugs into PFPE-based devices was drastically reduced compared to that into PDMS-based devices. The sorption into PFPE-based devices cannot be fully avoided for certain drugs; however, PFPE might be more useful in ADME evaluation than PDMS. Therefore, it would be important to choose the ‘right’ material for CIVM-based quantitative interpretation. Those advances in the development of intestine MPS using human intestinal organoids offer great potential for applications to pharmacokinetic research in drug development in a closer way to in vivo; however, at present, it is unclear as to what extent the system can be translated to in vivo (i.e., in vitro-in vivo extrapolation). Also, detailed characterization and comparison of those cutting-edge models with existing in vitro models such as human intestinal spheroid or organoid-derived monolayers and Caco-2 cells have been lacking, and further studies are required for the pharmaceutical industry to select and leverage the right models to use in drug discovery and development. 

In summary, the human intestinal organoid culture system is gaining interest in pharmacokinetic research as it better replicates the expressions and functions of pharmacokinetic-related factors in the human intestine compared to conventional models such as Caco-2 cells. This system overcomes the major limitations of the Ussing chamber system, such as the narrow time window for tissue viability and the limited availability of fresh human intestinal tissues, as intestinal organoids can be cryopreserved and used whenever needed. Although there are challenges in practical applications, such as the high cost and technical complexity of the maintenance of intestinal organoids, this system holds great promise for routine drug screening and candidate selection. It is necessary to accumulate data on inter-individual and inter-experimental variation in this system. Further studies are required to evaluate the system’s ability to assess intrinsic inter-individual variation in transport/metabolism activity; e.g., genetic polymorphisms of BCRP (*ABCG2* c.421C>A) [60]. Moreover, this system is expected to be a valuable tool for investigating species differences in intestinal drug absorption, similar to liver microsomes, because intestinal organoids can be generated from various animal species in an almost unified protocol [61]. Further studies on this aspect are needed.

## 4. iPSC-Derived Intestinal Epithelial Cells

Human iPSCs have emerged as a valuable tool for modeling the disposition of drugs in the human gut. Since the pioneering work of Spence et al. [40], iPSCs have been successfully differentiated into intestinal epithelial cells (IECs) using various protocols. However, it has long been an issue that the mRNA expression levels of drug-metabolizing enzymes and transporters, particularly P-gp and CYP3A4, are significantly lower in iPSC-derived IECs compared to the adult intestine [62,63,64].

Given those circumstances, Kabeya et al. demonstrated that the activation of the cAMP signaling pathway promotes the differentiation of iPSC-derived IECs. Treatment with 8-bromo-cyclic adenosine monophosphate (a cell permeable analog of cAMP) and 3-isobutyl-1-methylxanthine (a phosphodiesterase inhibitor) significantly enhanced the expression levels of genes related to pharmacokinetics [65]. Consequently, the expression levels of CYP2C9, CYP2C19, ABCB1/MDR1, and ABCG2/BCRP were higher in iPSC-derived IECs than in the adult small intestine. The metabolic activities of CYP2C9 and CYP2C19 were increased by 77.5-fold and 47.7-fold, respectively. Takayama et al. successfully generated functional monolayers of intestinal epithelial cells (IECM) through FOXA2 and CDX2 transduction, which resulted in enhanced gene expression levels of intestinal transporters (PEPT1, P-gp, and BCRP) and CYP3A4 [66]. The CYP3A4 activities in the transduced cells were significantly higher than those in control cells and Caco-2 cells. Furthermore, with this IECM model, they investigated the Fg value of midazolam by quantifying the concentrations of midazolam and its main metabolite, 1’-hydroxymidazolam, in the apical chamber, basal chamber, and cells. Consequently, the calculated in vitro Fg (0.73) closely matched the in vivo Fg (0.57). To accurately estimate intestinal metabolism, it would be advisable to consider CYP2C and UGT enzymes as well. Onozato et al. succeeded in building budding-like intestinal organoids with crypt-villus-like structures [67]. The expression level of PEPT1 was comparable to that in the adult small intestine; however, the expression levels of BCRP, P-gp, and CYP3A4 were lower than those in the small intestine. On the other hand, the organoid showed significant induction of P-gp and CYP3A4 by treatment with rifampicin and 1α,25-dihydroxyvitamin D_3_.

Considering the requirements of the pharmaceutical industry, accurate prediction of in vivo FaFg is of great importance, as Fa can be estimated from the apparent permeability in Caco-2 cells [68]. Akazawa et al. demonstrated a strong correlation between intrinsic clearance and in vivo parameters such as maximum plasma concentration (Cmax), area under the curve (AUC), and oral bioavailability (BA) using six prodrugs [69]. Moreover, the P_app_ values of 14 drugs, including three transporter substrates (P-gp and BCRP), exhibited a good correlation with their human Fa values. Similar approaches were adopted by Kabeya et al. [65], where forskolin-mediated cAMP signal activation resulted in higher gene expression levels of drug-metabolizing enzymes and transporters compared to those observed in the adult intestine. The membrane permeability was estimated using 16 compounds, including five CYP3A substrates, and a better correlation (r = 0.90) was observed between P_app_ and Fa or FaFg compared to Caco-2 (r = 0.56). However, CYP3A substrates were excluded due to deviation from the correlation curve, which might be attributed to the limited CYP3A4 activity in iPSC-derived IECs, presumably due to a smaller surface area of intestinal epithelial cells compared to the in vivo condition. Mayumi et al. proposed a novel approach by “boosting” CYP3A4 expression through vitamin D3/rifampin pre-treatment and applied it to extrapolate in vivo FaFg [70]. The predicted effective permeability values exhibited a good correlation with human FaFg values using 14 drugs, including CYP substrates (r^2^ = 0.8934). Furthermore, the obtained FaFg values were utilized to predict the apparent absorption rate constant and the pharmacokinetic profiles after oral administration of drugs using the advanced compartmental absorption and transit model. The absolute average fold errors of Cmax, Tmax, AUC, and BA were all less than 2, indicating the usefulness of their bottom-up approach for the pharmacokinetic prediction of orally administered drugs. iPSC-derived organoids are also desirable tools that have intrinsic intestinal properties; however, they have limitations in estimating permeability due to the opposite location of the brush border membrane.

Despite significant advancements in the physiological functions of iPSC-derived IECs over the past decade, further optimization is required to achieve quantitative prediction of intestinal drug disposition. MPS are expected to bridge this gap by enhancing the activities of drug-metabolizing enzymes and transporters through fluidic culture [71].

## 5. The Regulatory Approach towards CIVMs

This involves a careful consideration of their capabilities and limitations. CIVMs aim to replicate the physiological functions of human organs or tissues in a laboratory setting and can offer valuable insights into drug efficacy, toxicity, and disease mechanisms. As such, the regulatory authorities recognize the potential of these CIVMs to reduce the reliance on animal testing and provide more accurate and reliable data for decision-making. However, for CIVMs to be accepted by regulatory bodies generally, there remain challenges associated with validating and standardizing these CIVMs to ensure their reproducibility and reliability. To address these challenges, the regulatory approach emphasizes the need for a comprehensive understanding of the system’s characteristics, including its biological relevance, predictive capacity, and limitations. This involves evaluating the system’s ability to accurately represent human physiology, its performance in reproducing known responses to drugs or chemicals, and its reproducibility across different laboratories. In order to ensure the reliability and comparability of data generated from these CIVMs, the regulatory authorities encourage the development of standardized protocols and quality control measures. Additionally, they also emphasize the importance of transparent reporting and documentation of the system’s design, validation, and performance characteristics. 

More importantly, the regulatory approach promotes an iterative and collaborative process between the developers of these systems and the regulatory authorities, including engagement in early discussions to identify the regulatory expectations, providing guidance on study design and validation strategies, and facilitating the acceptance and adoption of these systems in regulatory decision-making processes. The regulatory approach towards CIVMs recognizes their potential in advancing regulatory science and decision-making, and it is essential to strike a balance between encouraging innovation and ensuring the reliability and reproducibility of data generated from these systems for regulatory purposes.

## 6. Conclusions

Recently, there has been much progress in the development of CIVMs that better mimic human organ structure, offering opportunities for their use in the field of ADME as well as pharmacology and toxicology studies. Due to the complex nature of the intestine, which acts as a gatekeeper for oral drug absorption, it is necessary to develop suitable in vitro system(s) to assess and predict intestinal absorption in humans. The most important consideration when selecting appropriate CIVMs for drug discovery and development is to recognize that each system has limitations, and it is crucial to define the CoU for each CIVM, as they serve specific purposes. It is no surprise that conventional Caco-2 cell monolayers are commonly used for high-throughput evaluation of drug permeability across the intestinal lumen, particularly for drugs with minimal or no involvement of drug metabolism. However, it is important to recognize that Caco-2 systems lack drug-metabolizing enzymes, which limits their utility in predicting the intestinal absorption of drugs that undergo extensive metabolism. The Ussing chamber system, which utilizes freshly isolated human intestinal tissue segments, effectively maintains the expression and function of drug-metabolizing enzymes and transporters, allowing for a more comprehensive evaluation of intestinal absorption. However, this system has limitations that hinder its widespread use in drug discovery and development, such as the relatively short viability of excised tissues and the need for multiple donor samples to account for inter-individual and inter-experimental variation, which limits its utility in routine drug screening. Recently, there has been increasing interest in the use of human intestinal spheroid/organoid or iPSC-derived cells, either in monolayer culture or MPS culture, for ADME research. These models are considered to be superior CIVMs as they better replicate the expressions and functions of pharmacokinetic-related factors in the human intestine compared to conventional models such as Caco-2 cells. Furthermore, this system overcomes the major limitations of the Ussing chamber system, which include the irregular availability of fresh human intestinal tissues and the limited viability of excised tissues. This provides great promise for the use of these models in drug discovery and development, as they are cost- and time-efficient solutions that better replicate organ functions in a manner closer to in vivo conditions. However, comprehensive data are still lacking, and further studies are needed to determine the extent to which each system can be translated to in vivo conditions based on detailed characterization and comparison. Based on these model-omics data, it is crucial to define the research questions for each project and select the appropriate CIVMs that are fit for purpose in drug discovery and development applications.

## Data Availability

Not applicable.

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
