# Peer review of "Recent Advances in the Gastrointestinal Complex in Vitro Model for ADME Studies"

_pharmaceutics, 2023, doi:10.3390/pharmaceutics16010037_

Round 1

Reviewer 1 Report

Comments and Suggestions for Authors

Reviewer’s comments

I was reviewing the REVIEW manuscript contribution ”Recent advances on gastrointestinal complex in vitro model for ADME studies” by Kazuyoshi Michiba, Kengo Watanabe, Tomoki Imaoka, and Daisuke Nakai to Pharmaceutics.

The authors outline the field of in-vitro testing of absorption and permeability, describe the most encountered challenges, potential pitfalls, and open for a perspective that novel complex in-vitro models (CIVM) may hold greater predictive potential. This review describes new complex in vitro gut models (gut CIVMs) , especially focussing on human intestinal 2D and 3D organoids derived from human biopsies or hiPScell derived enterocytes and compare them to the conventional golden standard assays Caco-2 and intestinal Ussing chambers highlighting advances made and major benefits of the gut CIVMs. There are several sub-sections in the body of the review addressing different in-vitro models and their main applications and limitations, and in a summative conclusion present their assessment of the current landscape.

The main merit of the paper is the strong conclusion and perspective that CIVM hold a lot of promise and now is the time to document and validate their real utility.

General comments:

It is a fairly short review which non the less gives a valuable overview of the gut CIVMs to come and the promise they hold for the future is richly exemplified.

There is a quite narrow focus on mostly Caco-2 cell monolayers as the conventional method to compare the new gut CIVMs against. There are ways to compensate for the lack of metabolic enzymes in the Caco-2 system through complementary data, the Fg may be predicted from HIMs or HLMs for example. When it comes to mechanistic studies of Fa, transfected cellines like MDCK are often used. As these models complement the Caco-2 studies it would have been good to mention their relevance among the conventional absorption models as well.  The authors may consider if the mention of low/absence of metabolism in Caco-2 cells needs repeating in multiple places in this review, it appears a bit repetitive on reading.

The successful absorption predictions of fresh (human) intestine in Ussing chambers that had been published (citations 20-23), however seem to be ruled out by the authors on quite light grounds (ease of access to material).  When discussing Complex IVM, are Ussing chambers generally more challenging to apply than CIVM?, could they be compared against CIVM in some places where only CAco-2 is discussed.

Would also have been nice to complement the review with some general pictures/figures of intestinal organoids and MPS systems as well as a general overview of the passage of a drug from the intestinal lumen into the blood with Fa and Fg marked out.

Specific comments in order of appearance in the text :

·         Page 1, line 14-16: sentence in abstract hard to understand –(“leverage” does not make sense here, some words appear to be missing)

·         Page 1, line 30: the sentence is describing intestinal absorption in general and by that I interpret it as describing the in vivo situation and which barriers need to be crossed, therefore I wonder why the serosal epithelial barrier is mentioned as it is not in scope in the in vivo setting.

·         Page 1, line 35-38: the text appears to report that Caco-2 cells are only useful to describe passive permeability, however were not useful to study transport and metabolism. Despite the high level of the introduction section, it must be acknowledged that Caco-2 cells are an established model to investigate efflux transport. The contribution of active transport to a compound’s absorption can be investigated using the Caco-2 as key intestinal transporters are present (e.g. Pgp, BCRP, PepT1, OATP2B1) albeit not with the exact same expression pattern as in human small intestine. This is indeed also supported by the reference 4 and it is also further on in the review (p6 line 302-303) stated that “..Fa can be estimated from the apparent permeability in Caco-2 cells” referring to ref 58. Please clarify those contradictory statements. Metabolism is present in Caco-2 cells however not a good representative of small intestinal metabolism (CYPs), but they express e.g. esterases.

·         Page 1, line 41-42: consider rephrasing “candidate selection of multiple compounds at once” – is it head-to-head comparison of potential candidates? (unclear)

·         Page 2, line 49: ”to realise precision medicine”   It is unclear where and why precision medicine for gastrointestinal absorption is required

·         Page 3 line 104-111, please comment on the fact (referring to ref 6) that no oxygenation was supplied to the excised tissue in the ice-cold buffer during the comparison of the 0 and 4h storage of the tissue before start of experiment and how that may have effected the reported transport decrease of the PEPT1 and PCFT substrate drugs and the transport increase of lucifer yellow.

·         Page 3 line 121; most Ussing chamber set ups nowadays house 8-12 parallel chambers, please comment more specifically why Ussing could not be used for candidate selection.

·         On page 3 and 4 the usefulness of intestinal organoids in studying metabolism and induction is exemplified which is interesting. However, the comparisons made to what Caco-2 can achieve when it comes to metabolism seems a bit short-stretching. It is a given that Caco-2 cells lack many of the metabolic enzymes, which is also already stated in the introduction. Worthwhile to compare to a system containing the metabolic enzymes (like human intestinal microsomes or human liver microsomes) and how these systems would compare to the intestinal organoids. Would for example intestinal organoids convey the intestinal metabolism more correctly than drawing conclusions on intestinal metabolism from a microsome based system?

General with the CIVM – could some statements about data on tightness and permeability be made, or are some models specifically useful to study metabolism however permeability and vectorial transport will be not possible to investigate as no tight monolayers are being formed? 

Comments on the Quality of English Language

The use of English language, grammar and typographic/spelling must be considered more carefully, it affects the text negatively and hampers readability and comprehendability.

·         Pluralis/singularis and present/past tense is intermixed

·         Articles (the) are frequently missing

·         prepositions are used in an odd manner

Some specific annotations:

·         The word “leverage” is used frequently also when more suitable words are available (p1, line 10 and 33)

·         Page 2, line 58-69, English needs to be corrected throughout this section.

·         Page 3 line 127; “compromised” should be changed to “comprised”

·         Page 7 line 326; “brash border membrane” should be changed to “brush border membrane”

Reviewer 2 Report

Comments and Suggestions for Authors

The review work “Recent advances on gastrointestinal complex in vitro model for ADME studies by Kazuyoshi Michiba and team ” is compilation gut complex in vitro models employing organoid intestinal model systems with and without MPS and describe gut complex in vitro models address research questions in ADME studies as compared with conventional method.

The minor comments are as follows:

1.       Additions of tabular summary of review work done.

2.       Illustration demonstrating key methods, model etc should be there to make it interesting for readers.

3.       Write about, Regulatory concerns if any

4.       Correct/add space between text and citation bracket. 

Reviewer 3 Report

Comments and Suggestions for Authors

The present review deals with the advances in complex in vitro models for the understanding of ADME profile of drugs in human gastrointestinal tract. The manuscript is well organized, the references cited are up-to-date. Although highly relevant for the field, a few major and minor revisions are necessary:

1. Line 35 "Caco-2 monolayers ...". It is not correct to state that Caco-2 model lacks metabolism and transport activities. It would be fair to state that Caco-2 cells lack the major drug metabolism enzyme CYP3A4, levels of other drug metabolising enzymes in Caco-2 cells are substantially high. Relevant drug transporters like P-gp, BCRP, OATP2B1 are also available in Caco-2 cells at sufficient levels. Nevertheless, the lack of CYP3A4 is the major drawback of using Caco-2 model to predict oral availability in human since many drugs are metabolized by CYP3A4.

2. Line 99 ", this system provided an appropriate estimation of ". It is not clear which system "this system" (Caco-2 or Ussing chamber) is referring to. 

3. Line 159 "...compared to Caco-2 cells[41]". The reference 5 which showed for the first time a comprehensive comparison between Caco-2 cells and a 2D intestinal model is missing here.

4. Line 224-252. In this section, the authors discussed the advances in organ chip research. A major aspect in this regard - the compound adsorption and absorption to the chip material and tubing systems which could make the ADME profiling impossible in such systems - is missing. 

5. Line 286 "Takayama et al. successfully...". The corresponding reference is missing.

Round 2

Reviewer 3 Report

Comments and Suggestions for Authors

All issues addressed. Great improvement after the revision.